# Harnessing Tumor-Infiltrating Lymphocytes in Triple-Negative Breast Cancer: Opportunities and Barriers to Clinical Integration

**DOI:** 10.3390/ijms26094292

**Published:** 2025-05-01

**Authors:** Cara Coleman, Tharakeswari Selvakumar, Aswani Thurlapati, Kevin Graf, Sushma Pavuluri, Shikhar Mehrotra, Ozgur Sahin, Abirami Sivapiragasam

**Affiliations:** 1Department of Hematology/Oncology, Hollings Cancer Center, Medical University of South Carolina, Charleston, SC 29425, USAselvakut@musc.edu (T.S.); pavuluri@musc.edu (S.P.); 2Department of Surgery, Medical University of South Carolina, Charleston, SC 29425, USA; 3Department of Biochemistry and Molecular Biology, Medical University of South Carolina, Charleston, SC 29425, USA; sahin@musc.edu

**Keywords:** triple-negative breast cancer, tumor infiltrating lymphocytes, tumor microenvironment

## Abstract

Triple-negative breast cancer (TNBC) continues to present a therapeutic challenge due to the fact that by definition, these cancer cells lack the expression of targetable receptors. Current treatment options include cytotoxic chemotherapy, antibody–drug conjugates (ADC), and the PD-1 checkpoint inhibitor, pembrolizumab. Due to high rates of recurrence, current guidelines for early-stage TNBC recommend either multi-agent chemotherapy or chemo–immunotherapy in all patients other than those with node-negative tumors < 0.5 cm. This approach can lead to significant long-term effects for TNBC survivors, driving a growing interest in de-escalating therapy where appropriate. Tumor infiltrating lymphocytes (TILs) represent a promising prognostic and predictive biomarker for TNBC. These diverse immune cells are present in the tumor microenvironment and within the tumor itself, and multiple retrospective studies have demonstrated that a higher number of TILs in early-stage TNBC portends a favorable prognosis. Research has also explored the potential of TIL scores to predict the response to immunotherapy. However, several barriers to the widespread use of TILs in clinical practice remain, including logistical and technical challenges with the scoring of TILs and lack of prospective trials to validate the trends seen in retrospective studies. This review will present the current understanding of the role of TILs in TNBC and discuss the future directions of TIL research.

## 1. Introduction

Subtypes of breast cancer are based on the tumor cell expression of molecular markers, including estrogen receptor (ER), progesterone receptor (PR), and human epidermal growth factor receptor 2 (HER 2). Triple-negative breast cancer (TNBC) is defined as tumors that do not express any of these three markers. Due to the absence of targetable surface receptors, chemotherapy was historically the mainstay of systemic treatment for TNBC. However, immunotherapy and antibody–drug conjugates (ADCs) have demonstrated significant efficacy in this subtype. Before the advent of these newer therapies, the progression-free survival (PFS) in stage IV TNBC after first-line treatment was as low as 3.9 months [1]. Moreover, survival following metastatic relapse remains shorter compared to other breast cancer subtypes, and the treatment options are more limited [2]. These challenges have led to an interest in the further characterization of the molecular biology underlying the heterogeneity and pathogenesis of TNBC, which can potentially lead to advances in treatment.

As demonstrated in other cancers, the tumor microenvironment (TME) of breast cancer plays an important role in tumor evolution and thus impacts disease progression. The TME is influenced by cell types, including tumor infiltrating lymphocytes (TILs), fibroblasts, and lymphatic vascular channels. Interestingly, higher proportions of TILs in the tumor microenvironment have been found to be associated with a better response to chemotherapy in TNBC [3,4]. In this context, recent research has focused on evaluating the prognostic and predictive value of TILs in patients with TNBC. This has led to extensive insights into the intricate interplay between TILs and tumor cells in impacting the TNBC disease course, while opening avenues for the further exploration of newer therapies, some of which this review aims to highlight.

## 2. The Role of TILs in the Tumor Microenvironment

### 2.1. Tumor Microenvironment (TME)

The TME plays a key role in the proliferation of cancer cells and is made up of immune cells, fibroblasts, stroma, and vasculature, which form an intricate network that contributes to both pro-tumor and antitumor effects. TILs are immune cells that exist within the TME and play an important role in modulating the immune response to cancer cell growth and proliferation. The immune cell types that are predominantly found in the TME are T and B lymphocytes, myeloid derived suppressor cells (MDSCs), tumor-associated macrophages, neutrophils, mast cells, natural killer cells, and dendritic cells [5].

The initial defense against emerging cancer cells is primarily mounted by antitumor immune cells, with CD8+ T-cells playing a leading role in cancer immunosurveillance. Initially, when the tumor develops, naïve T-cells undergo priming within the local lymph node or blood stream upon immunogenic antigen presentation by antigen-presenting cells (APCs). This is followed by their activation and migration to the tumor microenvironment [6]. Activated CD8+ T-cells differentiate into cytotoxic T lymphocytes (CTLs). These CTLs release cytolytic molecules, including perforin and granzyme, triggering an antitumor response that ultimately leads to the destruction of cancer cells. In addition to CD8+ T-cells, CD4+ T-cells contribute significantly to this immune response by secreting various cytokines. The CD4+ T lymphocytes are subdivided into T-helper 1 (Th1), Th2, Th9, Th17, and regulatory T-cell (Treg) groups [7,8]. Th1 CD4+ T-cells release pro-inflammatory signals, such as interferon-gamma (IFN-γ) and interleukin-2 (IL-2), which lead to the maturation of APCs and thus enhanced tumor antigen presentation to CD8+ T-cells. This results in the expansion of T-cells that are tumor-antigen-reactive. Furthermore, IL-2 promotes the maturation of CD8+ T-cells into fully functional CTLs. This coordinated effort among different types of immune cells leads to an antitumor immune response with specificity for tumor antigens, creating a natural defense against cancer development [7,9,10,11].

Cancer cells employ a range of sophisticated mechanisms to evade detection and destruction by the immune system, enabling their unchecked growth and proliferation. They can suppress the immune system by releasing chemokines such as CCL22 and CCL17, which bind to chemokine receptors on Tregs and recruit them to the tumor microenvironment [12]. Tregs suppress the antitumor function of the immune system by secreting the anti-inflammatory cytokines IL-10, TGF-β, and IL-35. Tregs also cause the direct destruction of cytotoxic T-cells via granzyme secretion [7]. In addition, Th2 CD4+ T-cells impact the tumor immune microenvironment via the secretion of IL-4, IL-5, IL-13, and IL-17, which exert a pro-tumor growth effect [13]. Tumor cells escape the immune system by downregulating the surface expression of proteins (such as MHC-1), rendering them undetectable to CTLs [14]. Tumor cells also lead to the upregulation of factors such as COX-2, PGE2, IL-6, and GMCSF within the TME, leading to immunosuppressive effects on T-cell activation. In addition, the secretion of growth factors, such as TGF-β and epidermal growth factor (EGF), can potentiate cancer cell proliferation and make them less susceptible to programmed cell death [7,15,16].

The dynamic interplay between these pro- and antitumor immune responses utilized by the diverse immune cells that comprise the TME governs the tumor growth potential and impacts cancer progression and metastasis (Figure 1).

### 2.2. TILs in Breast Cancer

In breast cancer, stromal TILs (sTILs) are present in the stromal region adjacent to the tumor margins and make up the majority of TILs. Intratumoral TILs (iTILs) are present within the tumor itself and make up a smaller proportion of TILs [4]. ITILs are formed from circulating lymphocytes that migrate across the tumor endothelial barrier to populate the tumor site. The tumor endothelium and stiff stroma around breast tumor cells are thought to prevent infiltration of TILs into the tumor, leading to the smaller proportion of iTILs [14]. The prognosis in breast cancer has been associated with both the number of TILs and the proportion of the specific types of TILs found in the lymphoid infiltrate surrounding the tumor [18]. Immune checkpoint ligands, such as PD-L1, on tumor cells can inhibit the activity of TILs, leading to immune evasion. In this context, TIL density in the TME may have a predictive value in terms of the response to PD1 checkpoint inhibitors [19]. TILs are found in primary breast cancers, as well as metastatic sites, but the density of TILs in metastatic sites tends to be lower when compared to the primary tumor sites. The impact of TILs in the tumor microenvironment of metastatic sites is yet to be fully characterized [20].

### 2.3. TILs in TNBC

TNBC is notable for its heightened immunogenicity relative to other breast cancer subtypes. This is reflected in the greater abundance of TILs within TNBC tumors. A positive correlation between the enrichment of certain types of TILs in the tumor microenvironment and favorable outcomes in TNBC has been reported [21]. Studies of TILS in TNBC have shown a median sTIL infiltration of between 15 and 25%, a median iTIL infiltration of between 5 and 10%, and a lymphocyte-predominant (LPBC) phenotype (presence of >50–60% TILs) prevalence of between 10 and 20% in primary tumors [22,23,24,25,26,27]. In comparison, luminal subtype breast cancers (expressing estrogen receptor and/or progesterone receptor) show a median infiltration of sTILs of between 7 and 10% [22,23,28] and of iTILs of between 1.5 and 5% [22,23,29], with a heterogenous prevalence of LPBC ranging from 2.9 to 12% [20]. In HER2-positive breast cancers, the median level of sTILs was noted to be between 15 and 20% [22,23,28,30,31], of median iTILs, it was between 3 and 10% [22,23,30], and the prevalence of the LPBC phenotype was between 10 and 11% [20,22,32]. The increased clonal heterogeneity, genomic instability, and higher mutational burden seen in TNBC are thought to lead to this increased immunogenicity compared to other subtypes [20,33].

The ratio of Tregs to CD8+ T-cells also differs between the subtypes of breast cancer. IHC staining for FOXP3, which is expressed by Tregs, can be used to quantify Tregs in breast cancer tissue. Papaioannou et al. found that TNBC and HER-2 positive tumors are associated with a higher Treg/CD8+ T-cell ratio compared to luminal subtypes. In addition, their study demonstrated that high Treg/CD8+ T-cell ratios were associated with higher grade tumors and higher Ki-67 expression [34].

The immune cell profiling of breast cancer tissue has demonstrated various predictive biomarkers, including host-related factors, immune-related cells, and genetic markers, showcasing the heterogeneity of TNBC. Burstein et al. have identified four subtypes of TNBC: Luminal Androgen Receptor (LAR), mesenchymal, basal-like immune-suppressed (BLIS), and basal-like immune-activated (BLIA) [35]. BLIA comprises the majority of TNBCs and has a complex genomic profile with a high frequency of deficiency in homologous recombinant mediated DNA repair (HRD), and more than 90% of these tumors harbor TP53 mutations. Interestingly this subset has also been found to have abundant TILs [36,37]. On the contrary, the BLIS subtype has increased TP53 mutations, but is associated with lower levels of TILs [36]. The mesenchymal subtype generally has low genomic complexity, the activation of the PI3K pathway, and very limited TILs [37,38]. The LAR subtype demonstrates low genomic complexity, low HRD, limited TILs, and mutations in PIK3CA, AKT1, NF1, GATA3, and CDH1 [39,40]. Notably, the BLIA subtype demonstrated the best disease-free survival (DFS) and disease-specific overall survival (DSS), and BLIS had the worst DFS and DFSS among the four subtypes [37].

## 3. Prognostic Value of TILs in TNBC

### 3.1. Early-Stage TNBC

In a 2016 meta-analysis, a higher level of TILs was associated with a better prognosis in TNBC [41]. Loi et al. demonstrated that each 10% increase in TILs was associated with a 13% decrease in the relative risk of distant recurrence in a study with 134 early-stage TNBC patients. The clinical benefits were seen regardless of the type of adjuvant chemotherapy used [28]. A German meta-analysis of 906 patients with early-stage TNBC showed that a 10% increase in TILs is associated with longer DFS and OS in TNBC. Their meta-analysis also demonstrated that TILs may be able to predict pCR rates in TNBC, with pCR seen in 31% in patients with low and intermediate TLS and 50% in high TILs [42]. The KEYNOTE 173 study showed that patients with a pCR after the combination of chemotherapy and pembrolizumab were more likely to have higher pretreatment sTILS levels (42% vs. 10%) [43]. A survival analysis of Colombian patients with TNBC showed longer OS, and a higher pCR was seen in patients with a higher infiltration of sTILs, and low levels of sTILS were associated with a higher mortality hazard [44]. Abdullaevo et al. concluded that the three-year event-free survival (EFS) in patients with high TIL levels was 95%, versus 65% in a low-TIL group (*p* = 0.037) [45]. The analysis of two phase III adjuvant breast cancer trials (ECOG 2197 and ECOG 1199) demonstrated that increasing sTILs predicted a lower risk of recurrence or death, distant recurrence, and overall mortality (Table 1). Their analysis also included iTILs and observed similar improved outcomes with higher iTILs, but since only 15% of the tumors had >10% iTILs, the results did not reach statistical significance [46]. In addition, the dynamics of TILs during neoadjuvant therapy may play an important role in predicting outcomes. A meta-analysis published in 2022 found that tumors with an increase in TILs post-neoadjuvant chemotherapy, compared to pre-chemotherapy, were associated with improved DFS in TNBC patients [47]. However, there are some exceptions. Some TNBCs, such as adenoid cystic carcinoma, are still considered to be a low risk, despite having low TILs [47]. Park et al. showed that sTILs can also identify a subset of stage 1 TNBC with an excellent prognosis, despite not receiving adjuvant chemotherapy [48]. Adding to this, in 2024 Guerts et al. discussed that patients with stage I or even stage II TNBC with high TILs have an outcome of >95% 5-year survival, despite not receiving systemic chemo–immunotherapy, recognizing them as low risk TNBC [48]. This is important to note, as most TNBC patients are regarded as high-clinical-risk, leading to the potential overtreatment of early-stage TNBC and the increased risk of toxicities.

Recent studies have suggested that the type of immune cells present in the TME, not just total TIL numbers, are important for influencing tumor growth and affecting prognoses. The presence of higher CD8+ T-cells, CD4+ T-cells, NK cells, and dendritic cells in the TME are associated with favorable outcomes in TNBC [54]. The single-cell profiling of breast cancer T-cells reveals that a rise in CD8+ T-cells corresponds to a rise in TILs [55]. The latest advances in profiling, utilizing single-cell RNA sequencing, demonstrated that a CD8+ T resident memory (Trm) gene signature was significantly associated with improved patient survival in early-stage TNBC and provided better prognostication than CD8 expression alone [55]. On the contrary, increased M2 macrophages and Tregs are associated with worse outcomes [56]. Early data suggest that a high ratio of cytotoxic T-cells (CTL) to Tregs is a strong predictor of achieving pCR in early-stage TNBC [57]. Onagi et al. demonstrated that TNBC tumors with high platelet-to-lymphocyte ratios (PLR) demonstrated more Tregs and higher PD-L1 expression in the tumor microenvironment. Patients with high TIL/PLR ratios demonstrated the lowest recurrence rate in the study (3.2%), compared to those with low TIL/PLR ratios, which had a recurrence rate of 28.6% [58].

Further categorizing CD8+ T-cells into functional and exhausted phenotypes may provide additional insights into the prognostic implications of these cells. When T-cells are exposed to chronic antigen stimulation, either due to chronic infections or cancer, they can develop into a distinct population of exhausted T-cells. These cells, when compared with effector or memory T-cells, exhibit decreased cytokine production, increased chemokine expression, and reduced proliferation when stimulated. They typically express inhibitory receptors, such as PD-1, TIM-3, LAG-3, CTLA-4, and TIGIT [59]. Studies have demonstrated that the characteristics of effector T-cells, such as higher T-cell receptor (TCR) clonality, higher PD-L1 positivity, and higher CD3:CD68 ratios, correlated with increased pCR [60]. In contrast, Cabioglu et al. found that high TIM-3 positivity (a characteristic of exhausted T-cells) on TILs from TNBC tumors was associated with a worse response to neoadjuvant chemotherapy [61].

Studies have also been conducted to assess any inter-racial and weight differences on the impact of TILs in TNBC. Even though TNBC is more common and aggressive in the African American population, studies demonstrate that there are no consistent racial differences in immune gene expression or TIL count in TNBC by race [62]. A population-based cohort study from the Northern California Breast Cancer Registry found no significant differences in sTIL scores among different racial and ethnic groups. However, higher continuous sTIL scores were associated with lower breast cancer-specific mortality, only in non-Hispanic White and Asian-American women, with no such association observed in African-American and Hispanic women. These findings highlight the complex factors influencing the treatment response and survival in breast cancer. While higher TIL levels typically indicate better prognoses, the study suggests that this relationship does not apply uniformly across all racial and ethnic groups [63]. Floris et al. investigated the effect of TILs and the body mass index (BMI) in patients with TNBC. They found that tumors with high TILs in lean patients have a higher pCR than in overweight individuals. Their study also demonstrated an increased EFS in lean patients with high TILs but not in heavier patients, but these findings were not statistically significant [64].

### 3.2. Metastatic TNBC

After determining the prognostic value of TILs in the early-stage TNBC setting, studies have tried to assess the role of TILs in the metastatic setting. CALGB 40502 was a randomized phase III trial of 799 patients with locally recurrent or metastatic breast cancer that compared bevacizumab plus either paclitaxel, nab-paclitaxel, or ixabepilone (Table 1). The Stover et al. [49] analysis of CALGB 40502 found lower numbers of TILs in distant metastatic sites when compared to primary tumors (8.4% vs. 13.3%, respectively). The authors noted that the analysis of TILs in lymph node tissue is less reliable given the presence of native nodal lymphocytes. Their study identified a higher proportion of hormone-negative tumors with high TILs (>5%) than hormone-positive tumors (64% vs. 33.8%, respectively). They found increased PFS in metastatic breast cancer with high TILs overall, but this was not statistically significant when the analysis was controlled for the hormone receptor status [49].

A retrospective analysis of metastatic lesions from 94 patients with metastatic breast cancer, including 43 TNBC and 51 HER2-positive, found low TIL levels overall, with a median of 5% in both subtypes. This supports the idea that TILs are significantly lower in metastatic sites. Only nine TNBC tumors were found to have TILs >10%, but within this subgroup, the median overall survival (OS) was found to be 62.9 months, compared to 11.8 months in the low TIL (<10%) subgroup. Of the TNBC samples, 29 also had CD8/FOXP3 ratios reported, and the study found that the highest tertile of the CD8/FOXP3 ratio was associated with improved median OS (54 months, *n* = 9), compared with the intermediate tertile (13.2 months, *n* = 10) and the lowest tertile (8 months, *n* = 9) [65].

## 4. TILs as Predictive Biomarkers for Response to Immunotherapy

Prior to the novel use of immunotherapy in the treatment of breast cancer, several studies identified the importance of TILs as a predictive marker for the response to chemotherapy in patients with TNBC. Similarly, an analysis put forth by the International TILs Working Group, published in 2015, identified early evidence that higher levels of sTILs found in tissue samples at diagnosis inferred better outcomes in the context of anthracycline-based chemotherapy [4]. More recently, immune checkpoint inhibitors (ICIs) have shown favorable results, with durable responses and improvements in both OS and PFS when compared to previous standardized therapies for TNBC [66]. Given the increasing utilization of ICIs, there has been significant research in identifying predictive markers for immunotherapy response.

As previously discussed, TILs represent a diverse range of lymphocyte subpopulations within the TME. While further studies are needed to fully explore the interaction between this heterogenous group of immune cells and immunotherapeutic treatments, the presence of CD4/8+ T-cells within the TME creates a plausible substrate for interaction with PD-1/PD-L1 inhibitors [67]. This known interaction between PD-1/PD-L1 inhibitors and several subsets of CD4/8+ T-cells serves as a likely initial step in understanding the immunologic, as well as the clinic, importance of TILs in the context of immunotherapy.

A recent study on combination neoadjuvant chemo–immunotherapy demonstrated a positive relationship between baseline sTIL levels and tumor response. In this phase 2 study involving 117 patients with stage I-III triple-negative breast cancer, who were treated with six cycles of carboplatin plus docetaxel and pembrolizumab, those with sTILs of 30% or greater achieved a pCR of 78%. In contrast, the pCR rate for patients with sTILs < 30% was 45% [68]. Perhaps most telling were the study design and results of the GeparNeuvo clinical trial, a randomized, phase II, double-blind, placebo-controlled study, which randomized early-stage TNBC patients to durvalumab or a placebo in addition to nab-paclitaxel, followed by standard epirubicin plus cyclophosphamide (Table 1). The patient randomization was stratified by the concentration of sTILs. The results showed that the patients with higher sTILs in both arms of the trial had significantly improved pCR rates (*p* < 0.01) and, overall, there was a trend for increased pCR rates in PD-L1-positive tumors [53]. The further translational analysis in the study assessed the predictive value of the tumor mutational burden (TMB) and TILs for pCR. The results, although potentially confounded by the simultaneous evaluation of TMB, showed the TIL concentration to be a component that independently predicted pCR. The groups that were identified to have high TMB and TIL concentrations were found to have an observed pCR of 82% (95% CI 60–95%), compared to only 28% (95% CI 16–43%) in the low-TMB and -TIL group [69]. These results confirmed the suspicion that TILs, in conjunction with immunotherapy, exert a more extensive antitumor effect.

The predictive utility of TILs has also been studied in the metastatic setting. A sub analysis of KEYNOTE-086, a phase II study of pembrolizumab monotherapy in metastatic TNBC, analyzed TIL levels in 193 patients (Table 1). The patients with increased levels of TILs (defined as >/= median amount within the sample of patients) showed increased overall response rates (odds ratio 1.26, 95% CI 1.03–1.55, *p* = 0.01) [51,70]. Furthermore, a sub-analysis of KEYNOTE-119 investigated the relationship between the presence of TILs and the outcomes in patients with metastatic TNBC receiving either pembrolizumab or chemotherapy (Table 1). The study was a negative trial that identified that TILs were significantly higher in responders and non-responders within the pembrolizumab arm but not in the chemotherapy arm. However, the median OS with pembrolizumab was 5.9 months for TILs <5% and 12.5 months for TILs >5%, compared to 8.8 and 11.3 months, respectively, with chemotherapy [52]. These correlations have extensive biologic plausibility, given that an increased immune presence at the tumor site would theoretically be strengthened by PD-1 blockade further enhancing the localized antitumor immune response.

Recently, in the adaptive BELLINI trial, 31 patients with early-stage TNBC were treated with neoadjuvant nivolumab with or without ipilimumab, eliminating neoadjuvant chemotherapy entirely (Table 1). A clinical response was observed in 38.7% of patients, with 10 patients achieving pCR. Only the patients with TILs ≥ 30% and PD-L1 ≥ 20% demonstrated clinical responses. The trial then opened cohort C, which included 15 patients with ≥ 50% TILs and node-negative disease. These patients underwent 6 weeks of nivolumab and ipilimumab followed by surgery. Five patients (33.3%) achieved pCR with this regimen of ICIs alone. This trial also utilized a spatial analysis and found that the responders had shorter distances from their CD8+ T-cells to the tumor cells. The non-responders were found to have higher levels of Treg cells post-treatment, supporting previous studies that have implicated Tregs in ICI resistance [50].

In addition to TNBC, TILs could also be utilized in tumors, like in high-risk luminal disease. These cancers with low ER levels behave aggressively, like TNBC [71]. Recently, it has been shown that high-risk luminal disease with high TILs responds very well to chemo–immunotherapy [72].

Overall TILs could be used in conjunction with a PD-L1 assay testing to determine the use of immunotherapy in TNBC. PD-L1 assays have variable sensitivities. For example, the SP 142-assay is known to be associated with false-negative results [73]. Given there is no level 1A evidence to support the use of TILs as predictive biomarkers yet, TILs may not replace or be utilized alone to support the use of immunotherapy but can be considered to work as an adjunct to the current biomarkers. However, if more prospective trials are conducted to validate the use of TILs as a predictive biomarker for the response to immunotherapy, they could make their way into clinical practice. This may be helpful, especially in developing countries, as a replacement to PD-L1 assays, which are expensive and only tested in limited centers.

## 5. Challenges

While some organizations now recommend the routine reporting of TILs on pathology reports, this has not been widely adopted as the standard practice. Currently, the European Society for Medical Oncology Early Breast Cancer guidelines and the World Health Organization recommend reporting TIL levels as a part of standard breast cancer pathology reports. The College of American Pathologists does not [74]. At the individual level, technical challenges exist with reporting due to lymphocyte heterogeneity and distribution, poor tissue sampling, and slide production issues. Significant work is underway from basic scientists, pathologists, and medical oncologists to develop clear tools for the measurement and reporting of TILs [75].

TILs are measured on the standard hematoxylin and eosin (H&E)-stained slides that are used to diagnose breast cancer, but pathologists need to undergo additional training in order to accurately assess and report TIL profiles. To address this issue, the international immuno-oncology working group has developed a training program to teach pathologists how to score TILs. Their program uses online videos and has allowed for an easy and economical method to increase the availability of pathologists who can score TILs. In 2017, the international immuno-oncology biomarkers working group proposed a standardized assessment method of TILs in invasive breast cancer [76]. However, the definition of which stromal tissue can be classified as peri-tumoral, and, therefore, be scored, has not yet been established by the guidelines in 2021 [77]. Also, counting and scoring TILs is dependent upon the skill and technique of each individual pathologist, and, therefore, some variation does exist. An evaluation of 32 trained pathologists demonstrated that the current method had intraclass correlation coefficients of between 0.81 and 0.93 for determining TILs above 30% and between 0.90 and 0.94 for determining TILs above 75% [78]. Ongoing efforts to develop a software-guided image assessment approach will further reduce intra-observer variation and help to standardize the process of TILs scoring [79].

There is currently no standardized threshold to distinguish between high and low TILs. Many studies use 30 or 50% as a cutoff when evaluating the prognosis as an outcome, while some studies evaluating the prediction of response to therapy use 5%. The international immuno-oncology working group recommends analyzing TILs as a continuous variable since there is a linear relationship between the TIL percentage and the prognosis. However, establishing a standardized threshold for low vs. high TILs would allow clinicians to more easily utilize the results of TIL measurements.

The presence of TILs in the tumor microenvironment is positively correlated with the presence of PD-L1 expression [80]. As discussed above, multiple trials have investigated TILs as a predictive marker for the response to immunotherapy. A systematic review in 2022 evaluated studies that investigated TILs as a predictive tool for the response to immunotherapy in multiple tumor types. Most of the available data were from retrospective analyses of clinical trials, and the review concluded that there was insufficient evidence to conclusively determine whether TIL profiles can predict the response to immunotherapy [81]. The specific cell types, level of activity, and spatial distribution (stromal vs. intratumor) of TILs are all important factors in the overall efficacy of the antitumor immune response [82]. These factors require additional investigation to determine the overall predictive effect of TILs on immunotherapy response.

## 6. Future Directions

Multiple recent retrospective studies have demonstrated the prognostic utility of TILs in early-stage TNBC. However, there have not yet been any large-scale prospective trials that validated TILs as a prognostic or predictive biomarker. While it has been hypothesized that certain patients with early-stage TNBC and high TILs could omit chemotherapy and still experience prolonged DFS, this idea has not yet been validated. Currently, patients with stage II or III TNBC are treated with a 6-month regimen of chemo–immunotherapy, which typically includes drugs such as doxorubicin, cyclophosphamide, carboplatin, paclitaxel, and pembrolizumab [83]. However, this regimen often results in significant toxicities, including neutropenic fever, cardiotoxicity, and neuropathy. Therefore, prospective studies are needed to safely explore treatment de-escalation strategies for patients who may achieve good outcomes with less intensive chemotherapy, thereby minimizing treatment-related adverse effects. One such trial is the ETNA study, which is evaluating whether patients with early-stage TNBC who have high sTILs can either de-intensify or omit chemotherapy safely. Following surgery, patients will be categorized into moderately high sTILs or high sTILs based on their sTIL percentage. The moderately high sTIL group will receive weekly paclitaxel and nine cycles of pembrolizumab. The high sTIL group will be followed on active surveillance with no adjuvant chemo–immunotherapy [84]. OPTimaL is another prospective study by the Netherlands cancer institute that is utilizing TILs as an integrated biomarker for guiding treatment in early-stage TNBC [85].

The type of T-cells present in the tumor microenvironment is important in understanding the role of TILs in breast cancer. The presence of Th1 CD4+ T-cells and cytotoxic CD8+ T-cells in the tumor microenvironment has been associated with an improved prognosis, while FOXP3+ regulatory T-cells are associated with the downregulation of the immune response and a poorer prognosis [80,86]. Because of these differences, future research that incorporates a determination of the type of immune cell, not just the total amount of TILs, may be able to more accurately predict prognoses and responses to immunotherapy. Spatial relationships between immune cells and tumor cells are also an important part of understanding TILs and their value as a biomarker [87]. New image-based, multiplexed platforms allow the mapping of cell–cell spatial relationships in tumor samples [88]. Spatial transcriptomics is a cutting-edge technology that allows the mapping of gene expression at a high resolution within tissue sections, preserving spatial information. This technology is particularly useful for understanding the TME and may be incorporated into future studies that examine the relationship between TILs and clinical outcomes [81]. In addition, image-based immune profiling techniques using multiplex IHC, mass cytometry, and single-cell RNA sequencing are able to assess the immune cell composition and the spatial distribution within the TME [89]. The Immunoscore (ISc) is a digital-pathology-based immunoassay that quantifies the densities of CD3+ and CD8+ T-cells within the tumor microenvironment. It has been validated internationally for its prognostic and predictive value in patients with stage III colon cancer [90]. The ISc analyzes both the location and the density of immune cells within the tumor microenvironment and quantifies specific immune cell subpopulations, such as cytotoxic CD8+ T-cells and memory T-cells. This is accomplished with immunohistochemistry and the multiplex immunofluorescence staining of immune cell biomarkers. Automated image analysis algorithms are then used to produce objective and reproducible data on the densities of different immune cell subpopulations [91]. The prognostic potential of the ISc has been evaluated in TNBC. A retrospective study performed the ISc process on tumors from 103 patients with early-stage breast cancer who received neoadjuvant chemotherapy, including 53 patients with TNBC. The study found that the pCR was significantly higher in patients with a high ISc (69%) than those with an intermediate ISc (50%) or a low ISc (6%) [92]. Various other studies have investigated adding a Treg analysis to the Immunoscore process to better characterize the competing pro- and anti-immunity mechanisms in play in the TME [93,94].

Advanced artificial intelligence (AI) techniques, such as deep learning, are being evaluated for their potential to overcome inter-observer variation. However, these models require enormous amounts of data and computing power to train, which has so far been prohibitive. Currently, work is underway to compile larger datasets for TILs in breast cancer, with annotated images that could be used to train AI models in TIL assessment. However, most TIL datasets include H&E images without IHC staining and, therefore, cannot be used to quantify subpopulations of TILs [95]. Future efforts to include IHC images in breast cancer TIL datasets could lead to advancements in our understanding of how these subpopulations of cells influence outcomes in TNBC.

In addition to establishing the prognostic and predictive value of TILs in TNBC, future research investigating the potential therapeutic potential of these cells is needed. In February 2024, the first adoptive cell transfer (ACT) utilizing TILs isolated from tumors was approved by the FDA for use in metastatic melanoma [96]. Compared with other types of ACT, TIL therapy has therapeutic advantages, including diverse T-cell receptor clonality, improved tumor infiltration, and reduced adverse effects [97]. Investigation into using adoptive TIL therapy in lung, head and neck, and cervical cancers is ongoing. A phase II clinical trial published in 2022 isolated TILs from resected lesions of 42 metastatic breast cancer patients. Eight of these were found to have appropriate reactivity, and enriched neoantigen-specific TILs were administered to six study patients, followed by a short course of pembrolizumab. Of these six patients, one experienced a durable complete response and two experienced partial responses [98]. Of note, this trial enrolled metastatic breast cancer patients regardless of their hormone receptor or HER-2 status. Given the current knowledge about the increased levels of TILs in TNBC tumors, it is possible that these results would be improved in trials that enrolled only TNBC patients. There is currently a clinical trial underway to further explore TIL therapy in triple-negative breast cancer. The TILS001 trial aims to explore the safety, tolerability, and efficacy of selected PD1+ T-cell infusions following prior selection based on high mRNA PD1 expression in patients with mTNBC. It is an open-label, single-arm, multicenter phase I/II prospective study with a two-stage design to evaluate treatment with PD1+ TIL infusions in advanced or mTNBC [99].

## 7. Conclusions

TILs are a pivotal component in the evolving landscape of triple-negative breast cancer, significantly affecting prognoses, treatment responses, and potential therapeutic strategies. Several studies have demonstrated the quantitative presence of TILs as an important prognostic factor in early-stage TNBC. These studies suggest that patients with early-stage TNBC and high TILs may benefit from de-escalated chemotherapy or even the omission of chemotherapy, while still achieving favorable outcomes. Studies have also indicated the predictive utility of TILs in response to immunotherapy, with improved responses and higher pCR rates being identified in tumors with high TILs. Although several retrospective studies have demonstrated this predictive utility of TILs in early-stage TNBC treated with immunotherapy, prospective trials, such as ETNA and OPTimaL, are currently evaluating their role as predictive biomarkers for the de-escalation of therapy in early-stage TNBC. Further prospective studies will assist to validate the role of TILs as a prognostic and predictive tool. The FDA has developed a course on TILs, which is the first ever course developed by the FDA on biomarkers and can be accessed on the TILs website at www.tilsinbreastcancer.org (accessed on 20 April 2025). In addition, future research investigating the therapeutic potential of TILs in TNBC is needed. Overall, TILs have level 1B evidence of their value as prognostic and predictive markers and can be considered for use together with other prognostic variables to estimate the risk of recurrence or the response to treatment in TNBC.

## Figures and Tables

**Figure 1 ijms-26-04292-f001:**
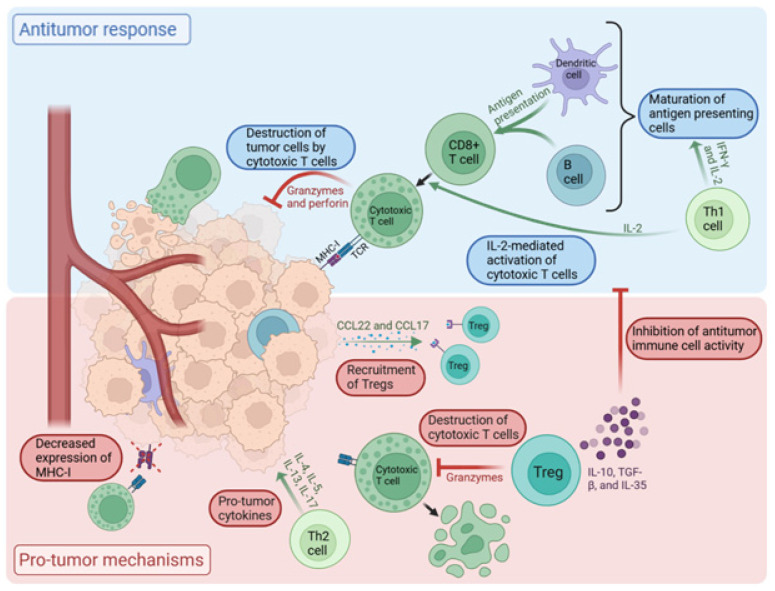
Pro- and antitumor mechanisms of TIL in the TME [17]. Antitumor responses are driven by Th1-mediated activation of CD8+ T-cells, which develop into tumor antigen-targeting CTLs. However, tumor cells employ immune evasion strategies, such as recruiting Tregs through chemokine release and downregulating MHC-I expression, to evade CTL detection. Pro-tumor cytokines released by Th2 further dampen antitumor immunity, allowing tumor cells to proliferate and metastasize. This intricate interplay governs the balance between immune surveillance and tumor escape, influencing cancer progression.

**Table 1 ijms-26-04292-t001:** A summary of clinical trials analyzing the impact of TILs on clinical outcomes in TNBC. Overall, higher sTIL levels were consistently associated with improved PFS and OS in both early-stage and metastatic TNBC, as well as an improved response to immunotherapy.

Trial	Design	Population	Interventions	Results
ECOG 2197 & 1199 (2014) [46]	Two phase III, randomized trials.	TIL analysis included 481 patients with operable TNBC (from both trials).	E2197: adjuvant doxorubicin plus either cyclophosphamide or docetaxel. E1199: adjuvant doxorubicin plus cyclophosphamide followed by one of four taxane regimens.	Higher sTIL score was significantly correlated with improved DFS, distant recurrence-free survival, and OS. Results of iTIL analysis did not reach statistical significance.
KEYNOTE-173 (2020) [43]	Phase Ib trial.	60 patients with early-stage TNBC.	Neoadjuvant pembrolizumab + chemotherapy.	Median pretreatment sTIL levels were higher in patients who achieved pCR than in those who did not (42% vs. 10%).
CALGB 40502 (2024) [49]	Phase III, randomized trial.	799 patients with advanced (stage IIIC or IV) breast cancer.	First-line nab-paclitaxel, ixabepilone, or paclitaxel, with or without bevacizumab.	Low sTIL scores were significantly associated with worse PFS (HR 1.34) and OS (HR 1.32). When controlled for hormone receptor status, the trend was similar but did not reach statistical significance.
BELLINI (2024) [50] Nederlof, I.; Isaeva, O.I.; de Graaf, M.; Gielen, R.C.A.M.; Bakker, N.A.M.; Rolfes, A.L.; Garner, H.; Boeckx, B.; Traets, J.J.H.; Mandjes, I.A.M.; et al. Neoadjuvant nivolumab or nivolumab plus ipilimumab in early-stage triple-negative breast cancer: A phase 2 adaptive trial. *Nat. Med.* **2024**, *30*, 3223–3235. https://doi.org/10.1038/s41591-024-03249-3. Available online: https://pubmed.ncbi.nlm.nih.gov/39284953/ (accessed on 16 March 2025).	Phase II, adaptive trial.	46 patients with early-stage TNBC.	Neoadjuvant nivolumab (with or without ipilimumab).	High pretreatment TILs were associated with improved response to immunotherapy. Responders had shorter CD8+ to tumor distance.
KEYNOTE-086 (2019) [51]	Phase II, single-arm trial.	254 patients with metastatic TNBC.	Pembrolizumab monotherapy after progression on one or more systemic therapies.	Patients with high sTILs (>/= median amount within the sample of patients) had increased ORR (odds ratio 1.26).
KEYNOTE-119 (2020) [52]	Phase III, randomized trial.	622 patients with previously treated metastatic TNBC.	Pembrolizumab monotherapy vs. single-agent chemotherapy.	High sTIL scores were significantly associated with improved OS, ORR, PFS, and duration of response in the pembrolizumab arm, but not the chemotherapy arm.
Gepar-Nuevo (2022) [53] 1. Loibl, S.; Untch, M.; Burchardi, N.; Huober, J.; Sinn, B.V.; Blohmer, J.-.; Grischke, E.-.; Furlanetto, J.; Tesch, H.; Hanusch, C.; et al. A randomised phase II study investigating durvalumab in addition to an anthracycline taxane-based neoadjuvant therapy in early triple-negative breast cancer: Clinical results and biomarker analysis of GeparNuevo study. Ann. Oncol. 2019, 30, 1279–1288. https://doi.org/10.1093/annonc/mdz158. Available online: https://pubmed.ncbi.nlm.nih.gov/31095287/ (accessed on 27 September 2024).	Phase II, randomized trial.	117 patients with metastatic TNBC.	Nab-paclitaxel followed by epirubicin and cyclophosphamide, plus durvalumab or placebo given every 4 weeks.	sTILs as a continuous variable were significantly associated with improved response rates in both treatment arms.

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
