# Peer review of "Harnessing Tumor-Infiltrating Lymphocytes in Triple-Negative Breast Cancer: Opportunities and Barriers to Clinical Integration"

_ijms, 2025, doi:10.3390/ijms26094292_

Round 1
Reviewer 1 Report
Comments and Suggestions for Authors
This is a very nice overview of the field of TILs and this manuscript is therefore certainly of value to the community. The following are additional nuances to make the manuscript more in line with ongoing discussions in the TIL-field:
- There should be a notion that not all TNBC are by default defined as clinically high-risk. There are two circumstances: 1. some TNBCs are considered as low-grade, such as for example adenoid cystic carcinoma, that are by default low-risk, even if they have low TILs (. doi: 10.1038/s41523-023-00554-x.) This should be mentioned somewhere. 2. If a patient with stage I or even stage II with high TILs has an outcome of >95% at 5-years, even if they are not treated with systemic chemotherapy, they should be considered as being of low-risk (please refer to this manuscript: doi: 10.1001/jamaoncol.2024.1917.). This is important to mention, as clinicians still consider a TNBC-patient as being of high clinical risk and there is a substantial overtreatment of early-stage TNBC due to this reason.
- It should be mentioned that TILs have level 1B evidence as a prognostic and predictive variable, and that in our daily practices we use prognostic biomarkers for decades already, all included in guidelines, that do not come even close to the evidence that there is on TILs, as nicely documented by the authors. For example, ONLY for some GEPs (mammaprint and oncotype-Dx) there is level 1A-evidence, and thus NOT for histological grade, prosigna, LVI, mitotic activity, etc…whilst these are often used in daily practices, for which never a prospective randomized design was ever developed. The solution to this, and this is why national and international guidelines are more and more often including TILs in their guidelines, is that TILs should be used TOGETHER with all other prognostic variables to estimate the risk of recurrence of a patient. LVI for example is not used as a binary variable to treat or not to treat, whilst only combined with all other variables it is useful. This notion should be more firmly established, as this supports the international tendency to include TILs in the reports of pathologists.
- Concerning prediction for immunotherapy, the general advice is that TILs should be analyzed together with PDL1, as a kind of quality control. We do know that PDL1-assays have variable sensitivities, such as the SP142-assay, which is notoriously know to be associated with false-negatives (doi: 10.1038/s41379-021-00884-w.), and if a case has high TILs every PDL1-assay SHOULD be positive, otherwise it is a false-negative PDL1-assay (doi: 10.1002/path.5406.). This notion is important to stress. The authors are right, there is no level1A-evidence for TILs as a predictive variable, and for prediction level 1A-evidence is preferred, so TILs should not be used to replace PDL1 in current practices. Yet, as the authors know, the costs of a PDL1-assay are prohibitive in some LMIC, so if a pathologist in say Bangladesh or certain parts of India sees that a patient with TNBC has very high levels of TILs, is it then not pragmatic to allow the TILs to select that patient for immunotherapy? Instead of performing a very expensive PDL1-assay? Moreover, the authors mention correctly the KN119-trial, but they need to specify that this was a NEGATIVE trial with PDL1 yet became positive using TILs. I invite the authors to reflect on all the above more and develop their own opinion.
- I would suggest to document briefly ongoing prospective trials using TILs, such as ETNA (see Pdf attached) and Optimal (NCT06476119), as examples of prospective trials that use TILs as an integral biomarker.
- The authors can add a line that CD8 TILs are the TILs, and that a rise in CD8 corresponds to a rise of TILs (doi: 10.1038/s41591-018-0078-7.).
- Recently, it was shown that high-risk luminal disease with high TILs respond very well to CT+CPI (doi: 10.1038/s41591-024-03414-8. ). It is becoming more evident that high-risk luminal disease, certainly with lower ER-values, behave like TNBC (doi: 10.1093/jnci/djae178.). This notion should be added, as this is driving the use of CPI in luminal disease.
- Line 340-341: "An ongoing challenge is that the definition of which stromal tissue can be classified as peri-tumoral and therefore be scored has not yet been established by the guidelines." This is partly correct. In the original guideline it was not indeed, yet in a subsequent reviews (doi: 10.1097/PAP.0000000000000161. and doi: 10.1097/PAP.0000000000000162.) it was.
- Line 372-375 " Prospective studies will be necessary in order for TILs assessment to become a prognostic tool that is useful in clinical practice. While it has been hypothesized that certain patients with early-stage TNBC and high TILs could omit chemotherapy and still experience prolonged DFS, this idea has not yet been validated [74]." This is partly true. As said before, currently, some clinicians use the TILs as an additional information on outcome that they use combined with all other prognostic variables). The TILs now considered to be for TNBC (and HER2+) what histological grade is for luminal disease. So, TILs are already being used in some daily practices, and the clinical trials mentioned above will provide clinicians that additional assurance they need on this biomarker, but to say that that the TILs are not ready for daily practice is exaggerated, as, again, we use prognostic factors in our daily practice, like LVI that have a much lower level of evidence than TILs.
- Finally, please add a link to the TILs-website (www.tilsinbreastcancer.org) as well as to the course the FDA has developed on TILs (the only course the FDA has ever developed on a biomarker), which can also be found on the TILs-website.

Author Response
Comments 1:- There should be a notion that not all TNBC are by default defined as clinically high-risk. There are two circumstances: 1. some TNBCs are considered as low-grade, such as for example adenoid cystic carcinoma, that are by default low-risk, even if they have low TILs (. doi: 10.1038/s41523-023-00554-x.) This should be mentioned somewhere. 2. If a patient with stage I or even stage II with high TILs has an outcome of >95% at 5-years, even if they are not treated with systemic chemotherapy, they should be considered as being of low-risk (please refer to this manuscript: doi: 10.1001/jamaoncol.2024.1917.). This is important to mention, as clinicians still consider a TNBC-patient as being of high clinical risk and there is a substantial overtreatment of early-stage TNBC due to this reason.
- Response 7: Thank you for this suggestion. We agree with the reviewer and incorporate this into our manuscript. This is mentioned under section 3.1 prognostic values of TILs in TNBC - early-stage breast cancer. This is mentioned along with the reference. This is noted on page 5 lines 208-210. All of this is highlighted in red.
- Response 8: Thank you for this suggestion. We agree with the reviewer and incorporate this into our manuscript. This is mentioned under section 4 TILs as predictive biomarkers for response to immunotherapy. These are on page 10 on lines 344-347. All of this is highlighted in red.
Reviewer 2 Report
Comments and Suggestions for Authors
This well-written and timely review covers the role of tumor-infiltrating lymphocytes (TILs) in triple-negative breast cancer (TNBC). The authors summarize TIL biology's current understanding in early-stage and metastatic TNBC and discuss their prognostic and predictive value, particularly in immunotherapy. Including recent clinical trials and emerging therapeutic strategies, such as adoptive TIL transfer, makes the review informative and relevant. I especially liked the cartoon in Figure 1—it’s visually appealing and helps convey the key concepts effectively.
I think the paper could be improved by adding a summary table of the major clinical trials mentioned (e.g., KEYNOTE-086, GeparNuevo, BELLINI), including trial design and key outcomes. This would give readers a clearer view of the clinical landscape and help organize some dense content. Also, while the discussion on immune cell subtypes and TIL heterogeneity is solid, the authors could briefly mention emerging technologies like spatial transcriptomics or image-based immune profiling, which are gaining traction and could be relevant for future studies. One specific suggestion: the legend for Figure 1 should be more detailed—readers should be able to understand the figure without reading the full review text for context.
There are a few minor points to fix. For example, “prognosticative” in line 52 might be better written as “prognostic.” Also, the section numbering is a bit inconsistent—there’s a 2.3.2 without a 2.3.1, which should be corrected. A quick check of figure placements and citation formatting would also help with clarity. Overall, this is a strong and informative manuscript with just a few areas that could be tightened up.
Author Response
Comments 1: I think the paper could be improved by adding a summary table of the major clinical trials mentioned (e.g., KEYNOTE-086, GeparNuevo, BELLINI), including trial design and key outcomes. This would give readers a clearer view of the clinical landscape and help organize some dense content.
Response 1: Thank you for this suggestion. We agree that adding a summary table of the major clinical trials will provide a clearer and more organized picture of the clinical landscape. Therefore, we have added a table summarizing the trial designs and key outcomes of the major clinical trials that were discussed in the manuscript. The table is included on page 8, line 339.
Comment 2:
While the discussion on immune cell subtypes and TIL heterogeneity is solid, the authors could briefly mention emerging technologies like spatial transcriptomics or image-based immune profiling, which are gaining traction and could be relevant for future studies.
Response 2:
Thank you for highlighting this. We agree that emerging technologies like spatial transcriptomics and image-based immune profiling are gaining importance in immuno-oncology research. We have added a brief description of these technologies in the "Future Directions" section on page 11, line 442. (Changes have been highlighted in red.)
Comment 3:
One specific suggestion: the legend for Figure 1 should be more detailed—readers should be able to understand the figure without reading the full review text for context.
Response 3:
Thank you for pointing this out. We have revised the legend for Figure 1 to make it more detailed and self-explanatory. The updated legend now includes additional context, which should allow readers to understand the figure without referring to the main text. (The updated figure legend is marked in red.)
Comment 4:
There are a few minor points to fix. For example, “prognosticative” in line 52 might be better written as “prognostic.” Also, the section numbering is a bit inconsistent—there’s a 2.3.2 without a 2.3.1, which should be corrected. A quick check of figure placements and citation formatting would also help with clarity.
Response 4:
Thank you for these helpful suggestions. We have made the following changes:
Corrected "prognosticative" to "prognostic" in line 52.
Fixed the section numbering to ensure consistency (2.3.2 has been changed to 2.3.).
Conducted a review of figure placements and citation formatting to improve clarity and ensure consistency throughout the manuscript. (All revisions are marked in red.)